# *Calcineurin B1* Deficiency Reduces Proliferation, Increases Apoptosis, and Alters Secretion in Enteric Glial Cells of Mouse Small Intestine in Culture

**DOI:** 10.3390/cells12141867

**Published:** 2023-07-17

**Authors:** Hikaru Teramoto, Naohide Hirashima, Masahiko Tanaka

**Affiliations:** Department of Cellular Biophysics, Graduate School of Pharmaceutical Sciences, Nagoya City University, Tanabe-dori, Mizuho-ku, Nagoya 467-8603, Japan

**Keywords:** enteric glial cell, small intestine, calcineurin, S100B, NF-κB, GFAP, GDNF, TGF-β1

## Abstract

To investigate the roles of calcineurin (CN) in glial cells, we previously generated conditional knockout (CKO) mice lacking *CNB1* in glial cells. Because these CKO mice showed dysfunction and inflammation of the small intestine in addition to growth impairment and postweaning death, we have focused on enteric glial cells (EGCs) in the small intestine. In this study, we examined the effects of *CNB1* deficiency on the proliferation and survival of EGCs and the expression and secretion of EGC-derived substances in culture to reveal the mechanisms of how *CNB1* deficiency leads to dysfunction and inflammation of the small intestine. In primary myenteric cultures of the small intestine, EGCs from the CKO mice showed reduced proliferation and increased apoptosis compared with EGCs from control mice. In purified EGC cultures from the CKO mice, Western blot analysis showed increased expression of S100B, iNOS, GFAP, and GDNF, and increased phosphorylation of NF-κB p65. In the supernatants of purified EGC cultures from the CKO mice, ELISA showed reduced secretion of TGF-β1. In contrast, GDNF secretion was not altered in purified EGC cultures from the CKO mice. Furthermore, treatment with an S100B inhibitor partially rescued the CKO mice from growth impairment and postweaning death in vivo. In conclusion, *CNB1* deficiency leads to reduced proliferation and increased apoptosis of EGCs and abnormal expression and secretion of EGC-derived substances, which may contribute to dysfunction and inflammation of the small intestine.

## 1. Introduction

Calcineurin (CN), a Ca^2+^/calmodulin-dependent protein phosphatase composed of A (catalytic) and B (regulatory) subunits, is involved in a number of intracellular signaling processes [1]. Although CN is involved in various aspects of neuronal development and function [2,3,4], its roles in glial cells remain to be elucidated [5,6]. Glial cells are a class of neural cells different from neurons and play important roles in the structure, function, and pathology of the nervous system [7,8,9].

To investigate the roles of CN in glial cells, we previously generated conditional knockout (CKO) mice lacking *CNB1* in glial cells by crossing floxed *CNB1* mice with *glial fibrillary acidic protein (GFAP)-Cre* mice [10]. In the enteric glial cells (EGCs) of these CKO mice, the nuclear factor of activated T cells, a CN-regulated transcription factor, failed to translocate into the nucleus after stimulation. Furthermore, the CKO mice showed mucosal degeneration and inflammation in the small intestine, reduced gastrointestinal motility, maldigestion and/or malabsorption, growth impairment in the weaning period, and eventual death after weaning [10,11]. These findings suggest that abnormalities in *CNB1*-deficiencient EGCs may be the primary cause of the phenotypes of the CKO mice. EGCs have been recognized as an important cell type constituting the enteric nervous system, which controls intestinal function and homeostasis by interacting with enteric neurons, epithelial cells, and immune cells via secreted substances [12,13,14,15,16]. However, it remains to be elucidated what the abnormalities in *CNB1*-deficiencient EGCs are in detail. 

In this study, therefore, we examined the effects of *CNB1* deficiency on the proliferation and survival of EGCs and the expression and secretion of EGC-derived substances in culture to reveal the mechanisms of how *CNB1* deficiency leads to dysfunction and inflammation of the small intestine. At the end, we examined the effects of S100 calcium-binding protein B (S100B) inhibition on growth impairment and postweaning death of the CKO mice in vivo because S100B was found to be upregulated in *CNB1*-deficiencient EGCs. The results obtained suggest that the abnormalities in *CNB1*-deficiencient EGCs may contribute to dysfunction and inflammation of the small intestine. 

## 2. Materials and Methods

### 2.1. Mice

*CNB1*-floxed (#006581) and *GFAP*-*Cre* (#004600) mice were purchased from Jackson Laboratory (Bar Harbor, ME, USA). The genotype of *CNB1*^fl/fl^ and *GFAP*-*Cre*^+/−^ is CKO. The other genotypes were used as controls. Animal experiments were carried out in accordance with the National Institutes of Health Guide for the Care and Use of Laboratory Animals (revised 1996) and approved by the Animal Care and Use Ethics Committee of Nagoya City University. All efforts were made to minimize the number of animals used and their suffering. 

### 2.2. Primary Myenteric Culture

The method for the primary culture of the myenteric plexus was described previously [10,17]. Muscle strips with the myenteric plexus were isolated in phosphate-buffered saline (PBS) by stripping the mucosa away from the small intestine of control and *CNB1*-CKO mice on postnatal day 20–22 using fine forceps. After cutting them into pieces approximately 2 mm in length with scalpel blades, the tissues were incubated for 1 h at 37 °C in 2 mL of a digestion solution composed of Eagle’s basal medium with Hank’s salts containing 300 U/mL collagenase (FUJIFILM Wako Pure Chemical, Osaka, Japan) and 20 U/mL deoxyribonuclease I (Sigma-Aldrich, St. Louis, MO, USA). After vortexing them for 15 s, the supernatant was collected and stored on ice. The undigested tissues were again subjected to incubation for 30 min in 1 mL of the digestion solution, vortexing, and collection of the supernatant. After the total supernatant was centrifuged (1100 rpm/220× *g*, 2 min, 4 °C), the cell pellet was suspended in 3 mL of a culture medium composed of Dulbecco’s modified Eagle’s medium/F12 (Gibco/Thermo Fisher Scientific, Waltham, MA, USA) supplemented with 10% fetal bovine serum (Sigma-Aldrich), 50 U/mL penicillin G potassium (Meiji Seika, Tokyo, Japan), and 100 µg/mL streptomycin sulfate (Meiji Seika). After centrifugation, the cell pellet was resuspended in 0.6 mL of the culture medium and strained through a 70-µm nylon mesh filter (Falcon/Corning, Corning, NY, USA), which produced a cell suspension with a concentration of 1–3 × 10^5^ cells per mL. Approximately 200 µL of the cell suspension was plated on a 35-mm culture dish (S35-DC12; Fine Plus International, Kyoto, Japan), the bottom of which was coated with poly-L-lysine (MW 30,000–70,000; Sigma-Aldrich). After 10 min incubation, 2 mL of the culture medium was added to each dish. The cells were incubated at 37 °C in 5% CO_2_/95% air. The culture medium was exchanged with fresh medium at 2, 6, 9, and 11 days in vitro (DIV). 

### 2.3. Proliferation Assay

Primary myenteric cultures were treated with 5-bromo-2’-deoxyuridine (BrdU; 20 μM; Sigma-Aldrich) for 18 h before fixation and fixed for 10 min in PBS containing 4% paraformaldehyde (FUJIFILM Wako Pure Chemical) at 3, 6, and 10 DIV. The fixed cells were pretreated for 20 min with 2 N HCl. After being blocked for 60 min at room temperature in PBS containing 2.5% normal donkey serum (Chemicon/Merck Millipore, Darmstadt, Germany) and 0.3% Triton X-100, they were incubated overnight at 4 °C in PBS containing rat anti-BrdU (1:1600; MAS 250c; Harlan Sera Lab, Loughborough, LE, UK), rabbit anti-GFAP (1:2400; 490740; Shandon, Pittsburgh, PA, USA), and mouse anti-alpha smooth muscle actin (αSMA) (1:1600; ab7817, Clone 1A4; Abcam, Cambridge, UK) antibodies. Subsequently, they were incubated overnight at 4 °C in PBS containing Cy3-conjugated anti-rat IgG (1:1600; 712-165-153), Cy2-conjugated anti-rabbit IgG (1:800; 711-225-152), and Cy5-conjugated anti-mouse IgG (1:600; 715-175-151) donkey antibodies (Jackson ImmunoResearch, West Grove, PA, USA). Finally, they were incubated for 30 min in PBS containing 0.25 μg/mL 4′,6-diamidino-2-phenylindole (DAPI; Sigma-Aldrich). Fluorescent images were acquired using a confocal laser scanning microscope (LSM800; Carl Zeiss, Jena, Germany) equipped with a 10× objective (N.A. = 0.45; Plan-Apochromat; Carl Zeiss). For quantification, we manually counted the number of GFAP- and BrdU-positive cells in images of 640 × 640 µm and calculated the ratio of GFAP- and BrdU-positive cells (proliferative EGCs) to GFAP-positive cells (total EGCs). The GFAP-positive cells were negative for αSMA. 

### 2.4. Cell Death Assay

Cell death assay was performed at 3, 6, and 10 DIV largely according to the manufacturer’s instructions. In brief, after being washed in binding buffer (10 mM Hepes (pH 7.4), 140 mM NaCl, 5 mM KCl, 1.8 mM CaCl_2_, 1 mM MgCl_2_), primary myenteric cultures were incubated for 15 min at room temperature in binding buffer containing Annexin V-FITC (1:10; Medical & Biological Laboratories, Nagoya, Japan), propidium iodide (PI) (5 μg/mL; Sigma-Aldrich), and Hoechst 33342 (5 μg/mL; Dojindo, Mashiki, Kumamoto, Japan). After the cultures were washed in binding buffer, fluorescent images were acquired using a confocal laser scanning microscope (LSM800). Subsequently, the cultures were fixed, processed for immunocytochemistry against GFAP using a Cy5-conjugated goat anti-rabbit IgG antibody (1:1600; 111-175-144; Jackson ImmunoResearch) as the secondary antibody, and imaged using a confocal laser scanning microscope (LSM800). For quantification, we manually counted the number of GFAP-, Annexin V-, and PI-positive cells in images of 640 × 640 µm and calculated the ratio of GFAP- and Annexin V-positive and PI-negative cells (early apoptotic EGCs) and GFAP-, Annexin V-, and PI-positive cells (late apoptotic/necrotic EGCs) to GFAP-positive cells (total EGCs). 

### 2.5. Purified EGC Culture

The method of purified EGC culture was described previously [17]. In brief, primary myenteric cultures were prepared, as described above, on 60-mm culture dishes. The culture medium was replaced with a serum-free culture medium composed of Dulbecco’s modified Eagle’s medium/F12 (Gibco/Thermo Fisher Scientific) supplemented with G5 (Gibco/Thermo Fisher Scientific) and N2 (FUJIFILM Wako Pure Chemical) supplements during 2–6 DIV. In addition, fibroblasts were scraped off with a 200-µL pipette tip (SATP-1002; Ikeda Scientific, Tokyo, Japan) at 2, 6, 9, and 11 DIV. This method generated a purity of approximately 90% EGCs. 

### 2.6. Western Blotting

Purified EGC cultures at 12–14 DIV were lysed in ice-cold suspension buffer (100 mM NaCl, 10 mM Tris-HCl (pH 7.6), 1 mM EDTA, 1 µg/mL aprotinin, 100 µg/mL phenylmethylsulfonyl fluoride), and an equal volume of 2× lysis buffer (100 mM Tris-HCl (pH 6.8), 4% sodium dodecyl sulfate (SDS), 10% 2-mercaptoethanol). The lysates were boiled for 10 min and sonicated for 30 s (Handy Sonic UR-20P; Tomy Seiko, Tokyo, Japan). After centrifugation, the supernatants were electrophoresed on 8 or 13% SDS-polyacrylamide gels. Proteins were transferred to nitrocellulose membranes (FUJIFILM Wako Pure Chemical) with an electroblotter. After blocking with nonfat dried milk, the membranes were probed with mouse anti-S100B (1:200; NBP2-53179, Clone 4C4.9; Novus Biologicals, Centennial, CO, USA), rabbit anti-nuclear factor-κB (NF-κB) p65 (1:1000; GTX102090; GeneTex, Irvine, CA, USA), rabbit anti-phospho-NF-κB p65 (Ser536) (1:2000; #3033, Clone 93H1; Cell Signaling Technology, Danvers, MA, USA), rabbit anti-inducible nitric oxide synthase (iNOS) (1:1000; GTX130246; GeneTex), rabbit anti-glial cell line-derived neurotrophic factor (GDNF) (1:200; #5098; BioVision, Milpitas, CA, USA), mouse anti-GFAP (1:4000; #270825, Clone G-A-5; Seikagaku, Tokyo, Japan), or mouse anti-β-actin (1:40000; A5441, Clone AC-15; Sigma-Aldrich) antibody. A horseradish peroxidase-conjugated goat anti-mouse IgG (1:1000; #330; Medical & Biological Laboratories) or anti-rabbit IgG (1:2000; #4050-05; Southern Biotech, Birmingham, AL, USA) antibody was used as the secondary antibody. The immunoreactivity was detected via enhanced chemiluminescence (Amersham ECL Western Blotting Analysis System or Amersham ECL Prime Western Blotting Detection Reagents; GE Healthcare, Little Chalfont, Buckinghamshire, UK) using a luminescent image analyzer (LAS-3000mini; FUJIFILM, Tokyo, Japan). For quantification, the band intensity was measured using the software program ImageJ Ver. 1.52a (National Institutes of Health, Bethesda, MA, USA), normalized by that of β-actin, and is shown as the ratio relative to the control. As for NF-κB p65, the band intensity of phospho-NF-κB p65 was normalized by that of NF-κB p65. 

### 2.7. ELISA

Supernatants of purified EGC cultures were collected at 12–14 DIV (3–5 d after the final medium exchange). The cells were lysed to determine the protein concentrations of each culture via the Pierce BCA Protein Assay Kit (Thermo Fisher Scientific). ELISA was performed using the Moues GDNF Rapid ELISA Kit (BEK-2229; Biosensis, Thebarton, Australia) or Mouse transforming growth factor-β1 (TGF-β1) DuoSet ELISA (DY1679; R&D Systems, Minneapolis, MN, USA), according to the manufacturers’ instructions. The samples for TGF-β1 ELISA were pretreated for 10 min with 0.17 N HCl to activate latent TGF-β1 to immunoreactive TGF-β1. The background concentrations of GDNF and TGF-β1 in the fresh medium were also determined and subtracted from the concentrations in the supernatants. GDNF and TGF-β1 concentrations were normalized by protein concentrations of each culture and are shown as ratios relative to the control. 

### 2.8. S100B Inhibition In Vivo

Pentamidine isethionate salt (Sigma-Aldrich) was dissolved in PBS at 1 mg/mL. A pair of control and *CNB1*-CKO mice were weaned on postnatal day 17.5 in each experiment. They were injected intraperitoneally once daily, starting on postnatal day 17.5, with pentamidine (10 mg/kg) or PBS [18]. 

### 2.9. Hematoxylin-Eosin Staining of Cryosections

The small intestine was dissected from pentamidine-administered control and *CNB1*-CKO mice on postnatal day 40, fixed for 45 min in PBS containing 4% paraformaldehyde (FUJIFILM Wako Pure Chemical), and sectioned at 12 µm on a cryostat (CM1850UV; Leica Microsystems, Wetzlar, Germany). The cryosections were stained with Mayer’s hematoxylin and 1% eosin Y solutions (FUJIFILM Wako Pure Chemical). Images were acquired using a transmission microscope (BX51; Olympus, Tokyo, Japan) equipped with a digital camera (DP72; Olympus). 

### 2.10. Statistical Analysis

Data are presented as mean ± SEM. Statistical analysis was performed using Student’s *t*-test. A value of *p* < 0.05 was considered statistically significant. 

**Figure 1 cells-12-01867-f001:**
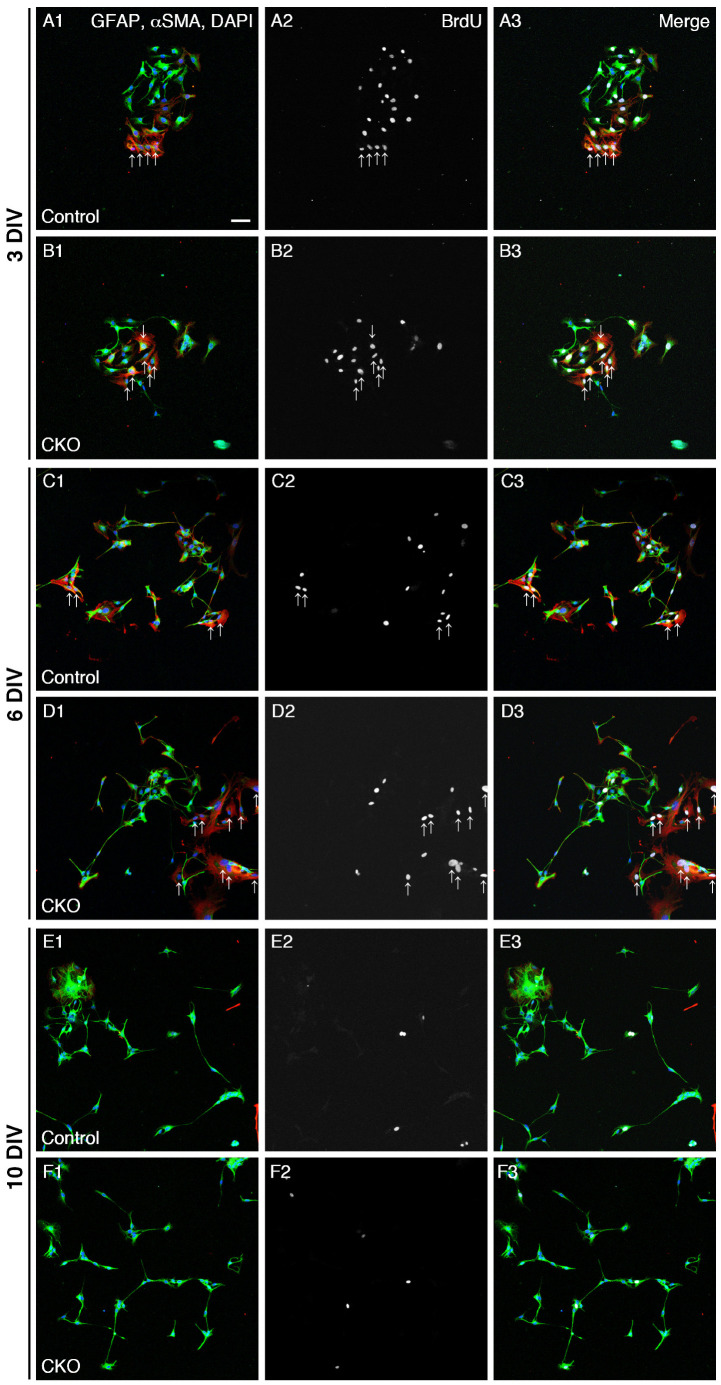
Reduced proliferation of *CNB1*-deficient EGCs. (**A**–**F**) Immunocytochemistry against BrdU (white), GFAP (EGC marker; green), and αSMA (fibroblast marker; red) and nuclear staining with DAPI (blue) of primary myenteric cultures from the small intestine of control (**A**,**C**,**E**) and *CNB1*-CKO (**B**,**D**,**F**) mice. The cultures were treated with 20 μM BrdU for 18 h before fixation and fixed at 3 (**A**,**B**), 6 (**C**,**D**), and 10 (**E**,**F**) days in vitro (DIV). Arrows indicate proliferative fibroblasts (αSMA+ and BrdU+ cells), which were excluded from the analysis. Scale bar in A1 (applies to all images) = 50 µm. (**G**–I) Ratios of proliferative EGCs (GFAP+ and BrdU+ cells to GFAP+ cells) in primary myenteric cultures from the small intestine of control and *CNB1*-CKO mice at 3 (**G**), 6 (**H**), and 10 (I) DIV. *n* = 125 ((**G**), control), 103 ((**G**), CKO), 106 ((**H**), control), 116 ((**H**), CKO), 112 ((**I**), control), and 99 ((**I**), CKO) images from three independent experiments. Error bars represent SEM. The ratio was significantly lower in *CNB1*-deficient EGCs than in control EGCs at 3 DIV (G; * *p* < 0.05, two-tailed unpaired *t*-test).

## 3. Results

### 3.1. Reduced Proliferation of CNB1-Deficient EGCs

As one aspect of the mechanisms of the dysfunction and inflammation of the small intestine in *CNB1*-CKO mice, we examined proliferation and survival of *CNB1*-deficient EGCs. For analysis of proliferation, primary cultures of the myenteric plexus from the small intestine of control and *CNB1*-CKO mice were treated with 20 μM BrdU for 18 h before fixation and fixed at 3, 6, and 10 DIV. The fixed cells were processed for immunocytochemistry against BrdU, GFAP (EGC marker), and αSMA (fibroblast marker) and nuclear staining with DAPI (Figure 1A–F), and ratios of GFAP+ and BrdU+ cells (proliferative EGCs) to GFAP+ cells (total EGCs) were calculated (Figure 1G–I). Immunocytochemistry against αSMA was performed in order to clearly distinguish EGCs from fibroblasts. The ratio of proliferative EGCs to total EGCs was significantly lower in *CNB1*-deficient EGCs than in control EGCs at 3 DIV, showing that proliferation of *CNB1*-deficient EGCs was reduced in the early culture period compared with control EGCs. 

### 3.2. Increased Apoptosis of CNB1-Deficient EGCs

Next, for analysis of survival, primary cultures of the myenteric plexus from the small intestine of control and *CNB1*-CKO mice were stained with Annexin V-FITC, PI, and Hoechst 33342 as a cell death assay at 3, 6, and 10 DIV (Figure 2A–F). After fluorescent images were acquired, they were fixed and processed for immunocytochemistry against GFAP, and the same optical fields were imaged again. Ratios of early apoptotic EGCs (GFAP+, Annexin V+, PI- cells to GFAP+ cells) and late apoptotic/necrotic EGCs (GFAP+, Annexin V+, PI+ cells to GFAP+ cells) were calculated (Figure 2G–L). Hoechst 33342 stains the nucleus of all of live and dead cells. As a result, ratios of early apoptotic EGCs were significantly higher in *CNB1*-deficient EGCs than in control EGCs at 3 and 6 DIV, showing that early apoptosis of *CNB1*-deficient EGCs was increased in the early and middle culture period compared with control EGCs. 

### 3.3. Increased Expression of S100B, iNOS, and GFAP and Phosphorylation of NF-κB p65 in CNB1-Deficient EGCs

As another aspect of the mechanisms of the dysfunction and inflammation of the small intestine in the CKO mice, we examined secreted substances from *CNB1*-deficient EGCs in culture. First, we focused on S100B, which is specifically secreted from EGCs in the intestine and induces neuronal damage and inflammation [12,19,20,21]. In this and the following analyses (Figure 3 and Figure 4), we used purified EGC cultures [17] because fibroblast contamination is problematic for analyses of secreted substances from EGCs in conventional primary myenteric cultures. Western blot analysis showed that expression of S100B was increased in purified EGC cultures from the CKO mice compared with control mice (Figure 3A). 

S100B activates the NF-κB pathway by binding to the receptor for advanced glycation end-products (RAGE) [12], leading to increased expression of NF-κB-regulated genes, including iNOS [12,20,22]. In addition, S100B itself is one of the NF-κB-regulated genes [12]. Therefore, we then examined phosphorylation of NF-κB and expression of iNOS. Western blot analyses showed that both phosphorylation of NF-κB p65 and expression of iNOS were increased in *CNB1*-deficient EGCs compared with control EGCs (Figure 3B,C). 

Thus, the NF-κB pathway was activated in *CNB1*-deficient EGCs. Because this is one aspect of the activation of glial cells [5,23], we examined expression of GFAP, a major marker of glial activation [8,23]. Western blot analysis showed that expression of GFAP was increased in *CNB1*-deficient EGCs compared with control EGCs (Figure 3D). 

### 3.4. Increased Expression but Normal Secretion of GDNF and Reduced Secretion of TGF-β1 in CNB1-Deficient EGCs

Next, we examined GDNF and TGF-β1 as other substances secreted from EGCs. GDNF [24,25,26] and TGF-β1 [27,28,29] enhance intestinal epithelial barrier maturation and wound healing via the regulation of proliferation and differentiation and the upregulation of intercellular junctional proteins in epithelial cells. In addition, GDNF [24,30] and TGF-β1 [27,31,32] have anti-apoptotic and restitution effects in epithelial cells, respectively. Western blot analysis showed that expression of GDNF was increased in purified EGC cultures from the CKO mice compared with control mice (Figure 4A). However, ELISA using the supernatants of purified EGC cultures showed that GDNF secretion was not altered in *CNB1*-deficient EGCs compared with control EGCs (Figure 4B). In contrast, TGF-β1 secretion was reduced in *CNB1*-deficient EGCs compared with control EGCs (Figure 4C). 

### 3.5. Effects of S100B Inhibition on CNB1-CKO Mice In Vivo

Finally, we examined the effects of S100B inhibition on growth impairment and postweaning death of the CKO mice in vivo. Intraperitoneal administration of 10 mg/kg pentamidine, an S100B inhibitor [33,34,35,36,37], partially rescued the CKO mice from postweaning death (Figure 5A). Intraperitoneal administration of PBS did not rescue the CKO mice from postweaning death (Figure 5A) or growth impairment (Figure 5B). In contrast, the growth partially recovered in pentamidine-administered CKO mice that survived beyond postnatal day 40 (Figure 5C), but not in pentamidine-administered CKO mice that died before postnatal day 25 (Figure 5B). Although repeated-measures ANOVA indicated that body weight was still significantly different between the control and CKO mice, the partial growth recovery was obviously observed in Figure 5C. Further, histological analysis via hematoxylin-eosin staining showed that the small intestine of pentamidine-administered CKO mice on postnatal day 40 was apparently normal (Figure 5D,E). 

## 4. Discussion

In the present study, *CNB1*-deficient EGCs showed reduced proliferation and increased apoptosis compared with control EGCs (Figure 1 and Figure 2). Because EGCs are an important cell type constituting the enteric nervous system that controls intestinal function and homeostasis [12,13,14,15,16], reduced proliferation and increased apoptosis of EGCs are potential causes of the dysfunction and inflammation of the small intestine in *CNB1*-CKO mice [10,11]. 

Although CN is known to play a role in apoptosis in several types of cells, especially in lymphocytes [1], it has also been reported that CN plays an anti-apoptotic role in T cells and fibroblasts [38,39]. Our result also suggests an anti-apoptotic role of CN in EGCs. Thus, CN may play a dual role in the regulation of apoptosis. 

On the other hand, *CNB1*-deficient EGCs themselves showed several abnormal properties (Figure 6). First, *CNB1*-deficient EGCs were activated in that GFAP expression and NF-κB p65 phosphorylation were increased (Figure 3). Concurrently, however, they were inactivated in that proliferation and survival were reduced, as described above. Showing such conflicting aspects is a characteristic nature of *CNB1*-deficient EGCs. 

EGCs interact with enteric neurons, epithelial cells, and immune cells through a variety of secreted substances [12,13,14,15,16]. As a result of analyses of alterations in the secreted substances from *CNB1*-deficient EGCs, we found that S100B expression was increased (Figure 3). The increased levels of S100B might induce neuronal damage and inflammation in the small intestine of *CNB1*-CKO mice. Indeed, treatment with an S100B inhibitor partially rescued *CNB1*-CKO mice from growth impairment and postweaning death in vivo (Figure 5). These results suggest an important contribution of S100B to the dysfunction and inflammation of the small intestine of *CNB1*-CKO mice. 

Because S100B activates the NF-κB pathway through its receptor RAGE [12], increased expression and secretion of S100B would lead to NF-κB activation in *CNB1*-deficient EGCs. The result that NF-κB p65 phosphorylation was increased is consistent with this. NF-κB activation would result in increased production of various inflammatory cytokines such as tumor necrosis factor-α, interleukin-1, interleukin-6, and cyclooxygenase-2 downstream of the NF-κB pathway (Figure 6), although these changes remain to be elucidated. Analyses of these changes are one of the important issues for a future study. 

In addition, NF-κB also upregulates iNOS [12,20,22]. In fact, iNOS expression was increased in *CNB1*-deficient EGCs (Figure 3). The increased expression of iNOS, leading to increased release of nitric oxide, may also contribute to inflammation in the small intestine of *CNB1*-CKO mice (Figure 6). 

On the other hand, NF-κB also upregulates S100B [12]. Thus, a positive feedback regulation loop of S100B-NF-κB signaling would be formed to drive EGCs to an activated state, leading to exacerbation of the dysfunction and inflammation of the small intestine of *CNB1*-CKO mice (Figure 6). 

As another secreted substance from EGCs, GDNF was upregulated in expression in *CNB1*-deficient EGCs (Figure 4). It is known that GDNF is upregulated in inflammatory bowel disease and could act to protect intestinal epithelial cells from cytokine-induced apoptosis [30,40]. Our result suggests that CN may suppress expression of GDNF under normal conditions, or that increased GDNF expression occurred as a compensatory response to small intestine inflammation in *CNB1*-CKO mice. However, secretion of GDNF was not altered in *CNB1*-deficient EGCs (Figure 4). The discrepancy between increased expression and normal secretion could be explained by hypothesizing that CN is also involved in a secretory mechanism of GDNF. If the CN-mediated secretory mechanism of GDNF was impaired because of CN deficiency, GDNF secretion might not be altered regardless of its increased expression. Exploring such a CN-mediated secretory mechanism is an interesting issue for a future study. 

In addition, TGF-β1 secretion was reduced in *CNB1*-deficient EGCs (Figure 4). The reduced secretion of TGF-β1, together with activation of the S100B-NF-κB signaling pathway, may induce inflammation in the small intestine of *CNB1*-CKO mice because TGF-β1 is critical to intestinal epithelial barrier maturation and wound healing [27,28,29]. 

Finally, pentamidine treatment partially rescued *CNB1*-CKO mice from growth impairment and postweaning death in vivo (Figure 5). Although pentamidine is primarily known as an antiprotozoal drug, it has been found that this drug also inhibits the NMDA receptor [41,42] and S100B activity [33,34]. Thereafter, pentamidine has been used in S100B-targeted treatment of melanoma [43] and S100B inhibition experiments in the pathological intestine and brain in vivo [18,35,36,37,44]. Although our previous study demonstrated that the CKO mice showed histological abnormalities in the small intestine [10], no apparent abnormalities were observed in the small intestine of pentamidine-administered CKO mice that survived to postnatal day 40 in the present study. These results, together with the increased levels of S100B in *CNB1*-deficient EGCs, suggest that upregulated S100B may induce abnormalities in the small intestine and result in growth impairment and postweaning death of the CKO mice. 

In conclusion, *CNB1* deficiency leads to reduced proliferation and increased apoptosis of EGCs and abnormal expression and secretion of EGC-derived substances (Figure 6), which may contribute to dysfunction and inflammation of the small intestine. 

## Figures and Tables

**Figure 2 cells-12-01867-f002:**
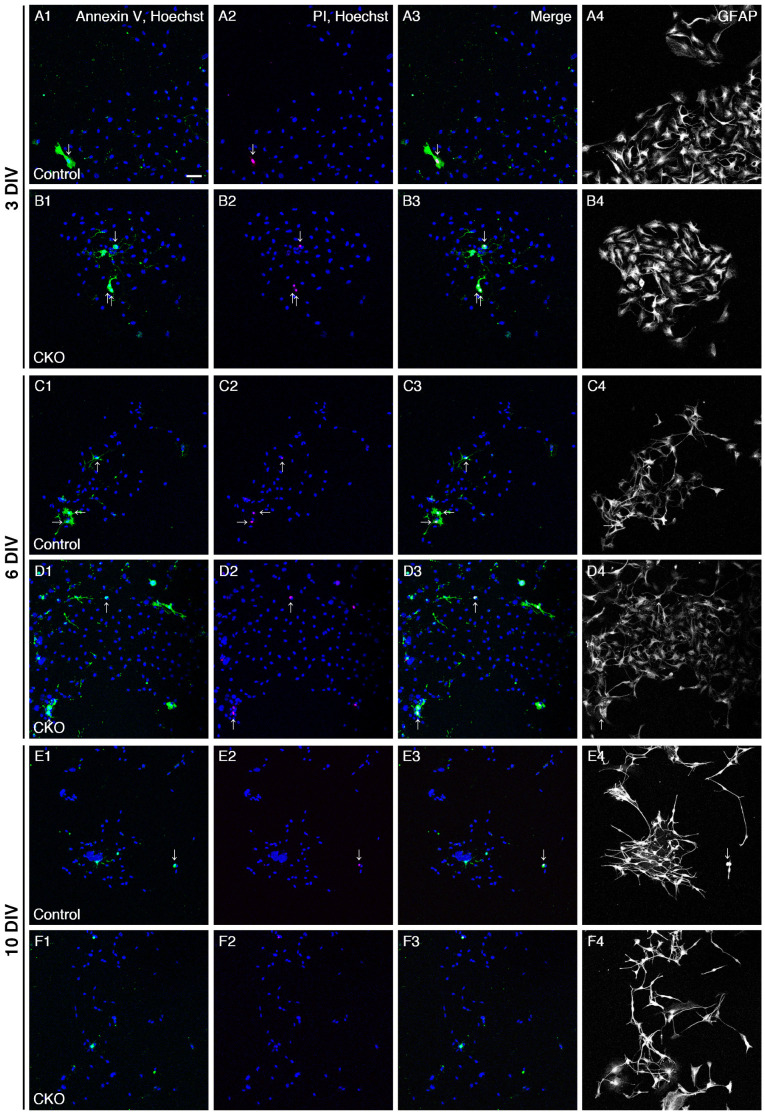
Increased apoptosis of *CNB1*-deficient EGCs. (**A**–**F**) Live staining with Annexin V-FITC (green), PI (red), and Hoechst 33342 (blue) and immunocytochemistry against GFAP (white) of primary myenteric cultures from the small intestine of control (**A**,**C**,**E**) and *CNB1*-CKO (**B**,**D**,**F**) mice. The cultures were stained with Annexin V-FITC, PI, and Hoechst 33342 as a cell death assay at 3 (**A**,**B**), 6 (**C**,**D**), and 10 (**E**,**F**) DIV. Immediately after fluorescent images were acquired, they were fixed and processed for immunocytochemistry against GFAP, and the same optical fields were imaged again. Arrows indicate late apoptotic/necrotic EGCs (GFAP+, Annexin V+, PI+ cells). Scale bar in A1 (applies to all images) = 50 µm. (**G**–**L**) Ratios of early apoptotic EGCs (GFAP+, Annexin V+, PI- cells to GFAP+ cells) and late apoptotic/necrotic EGCs (GFAP+, Annexin V+, PI+ cells to GFAP+ cells) in primary myenteric cultures from the small intestine of control and *CNB1*-CKO mice at 3 (**G**,**H**), 6 (**I**,**J**), and 10 (**K**,**L**) DIV. *n* = 110 ((**G**,**H**); control), 109 ((**G**,**H**); CKO), 82 ((**I**,**J**); control), 97 ((**I**,**J**); CKO), 75 ((**K**,**L**); control), and 82 ((**K**,**L**); CKO) images from three independent experiments. Error bars represent SEM. Ratios of early apoptotic EGCs were significantly higher in *CNB1*-deficient EGCs than in control EGCs at 3 and 6 DIV (G, I; * *p* < 0.05, two-tailed unpaired *t*-test).

**Figure 3 cells-12-01867-f003:**
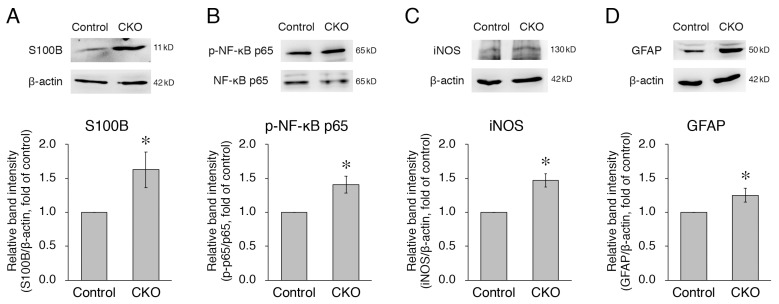
Increases in expression of S100B, iNOS, and GFAP and phosphorylation of NF-κB p65 in *CNB1*-deficient EGCs. (**A**,**C**,**D**) Representative Western blots of S100B (**A**), iNOS (**C**), GFAP (**D**), and β-actin (**A**,**C**,**D**) and quantification of the band intensity of S100B (**A**), iNOS (**C**), and GFAP (**D**) for purified EGC cultures from the small intestine of control and *CNB1*-CKO mice. For quantification, the band intensity of each protein was normalized by that of β-actin and is shown as the ratio relative to the control. *n* = 8 (**A**), 3 (**C**), and 11 (**D**) pairs of control and *CNB1*-deficient EGCs. Error bars represent SEM. The band intensities of S100B (**A**), iNOS (**C**), and GFAP (**D**) were significantly stronger for *CNB1*-deficient EGCs than for control EGCs (* *p* < 0.05, two-tailed paired *t*-test). (**B**) Representative Western blots of NF-κB p65 and phospho-NF-κB p65 and quantification of the band intensity of phospho-NF-κB p65/NF-κB p65 for purified EGC cultures from the small intestine of control and *CNB1*-CKO mice. For quantification, the band intensity of phospho-NF-κB p65 was normalized by that of NF-κB p65 and is shown as the ratio relative to the control. *n* = 5 pairs of control and *CNB1*-deficient EGCs. Error bars represent SEM. The band intensity of phospho-NF-κB p65/NF-κB p65 was significantly stronger for *CNB1*-deficient EGCs than for control EGCs (* *p* < 0.05, two-tailed paired *t*-test).

**Figure 4 cells-12-01867-f004:**
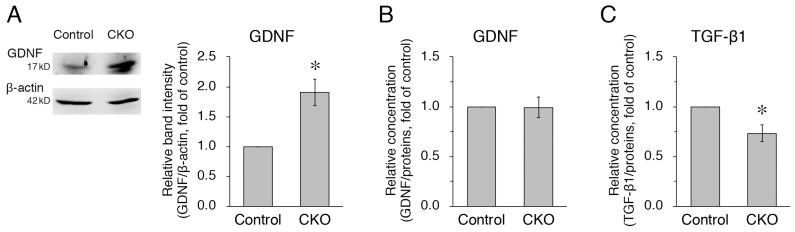
Increased expression but normal secretion of GDNF and reduced secretion of TGF-β1 in *CNB1*-deficient EGCs. (**A**) Representative Western blots of GDNF and β-actin and quantification of the band intensity of GDNF for purified EGC cultures from the small intestine of control and *CNB1*-CKO mice. For quantification, the band intensity of GDNF was normalized by that of β-actin and is shown as the ratio relative to the control. *n* = 7 pairs of control and *CNB1*-deficient EGCs. Error bars represent SEM. The band intensity of GDNF was significantly stronger for *CNB1*-deficient EGCs than for control EGCs (* *p* < 0.05, two-tailed paired *t*-test). (**B**,**C**) ELISA analysis of GDNF (**B**) and TGF-β1 (**C**) concentrations in the supernatants of purified EGC cultures from the small intestine of control and *CNB1*-CKO mice. The concentration of each protein was normalized by a protein concentration of each culture and is shown as the ratio relative to the control. *n* = 5 (**B**) and 6 (**C**) pairs of control and *CNB1*-deficient EGCs. Error bars represent SEM. The GDNF concentration was not different between control and *CNB1*-deficient EGC supernatants (**B**). The TGF-β1 concentration was significantly lower in *CNB1*-deficient EGC supernatants than in control EGC supernatants (* *p* < 0.05, two-tailed paired *t*-test) (**C**).

**Figure 5 cells-12-01867-f005:**
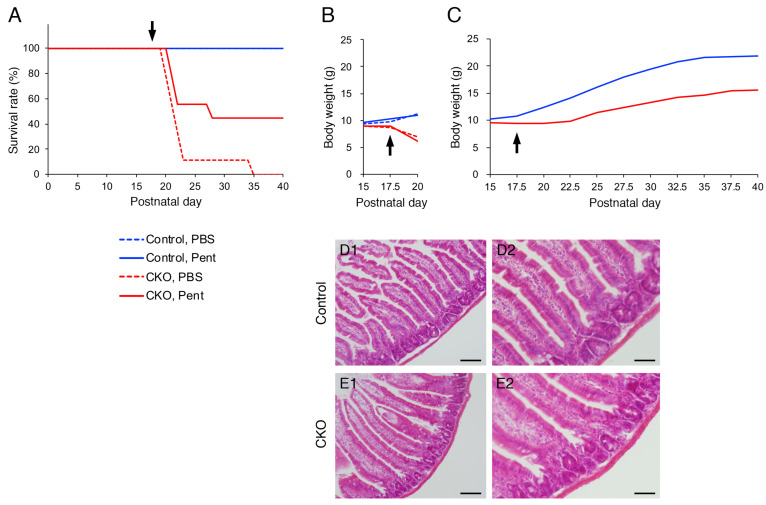
Effects of an S100B inhibitor on *CNB1*-CKO mice in vivo. (**A**) Survival curves of control (blue) and *CNB1*-CKO (red) mice into which pentamidine (10 mg/kg; solid lines) or PBS (dotted lines) was intraperitoneally administered from postnatal day 17.5 (arrow). *n* = 9 pairs of control and CKO mice. Solid blue line (control, pentamidine) and dotted blue line (control, PBS) overlap. The survival rate was higher in pentamidine-administered CKO mice (solid red line) than in PBS-administered CKO mice (dotted red line) during postnatal day 20–40. (**B**) Growth curves of control (blue) and *CNB1*-CKO (red) mice administered pentamidine (solid lines) or PBS (dotted lines). Data from experiments in which the CKO mice died before postnatal day 25 are shown. As for PBS administration, one CKO mouse among the 9 pairs died on postnatal day 35; thus, the data from the pair including this mouse were not included (*n* = 8 pairs). Growth did not recover in PBS-administered CKO mice. As for pentamidine administration, one CKO mouse among the 9 pairs died on postnatal day 28; thus, the data from the pair including this mouse were not included (*n* = 4 pairs). The growth did not recover in pentamidine-administered CKO mice that died before postnatal day 25. (**C**) Growth curves of control (blue) and *CNB1*-CKO (red) mice administered pentamidine. Data from experiments in which pentamidine-administered CKO mice survived beyond postnatal day 40 are shown. *n* = 4 pairs. Although repeated-measures ANOVA indicated that body weight was still significantly different between the control and CKO mice (*p* < 0.005), growth partially recovered in pentamidine-administered CKO mice that survived beyond postnatal day 40. (**D**,**E**) Hematoxylin-eosin staining of cryosections of the small intestine of pentamidine-administered control (**D**) and *CNB1*-CKO (**E**) mice on postnatal day 40. Low-power (**D1**,**E1**) and high-power (**D2**,**E2**) views. The small intestine of pentamidine-administered CKO mouse was apparently normal (**E**). Scale bars = 100 (**D1**,**E1**) or 50 (**D2**,**E2**) µm.

**Figure 6 cells-12-01867-f006:**
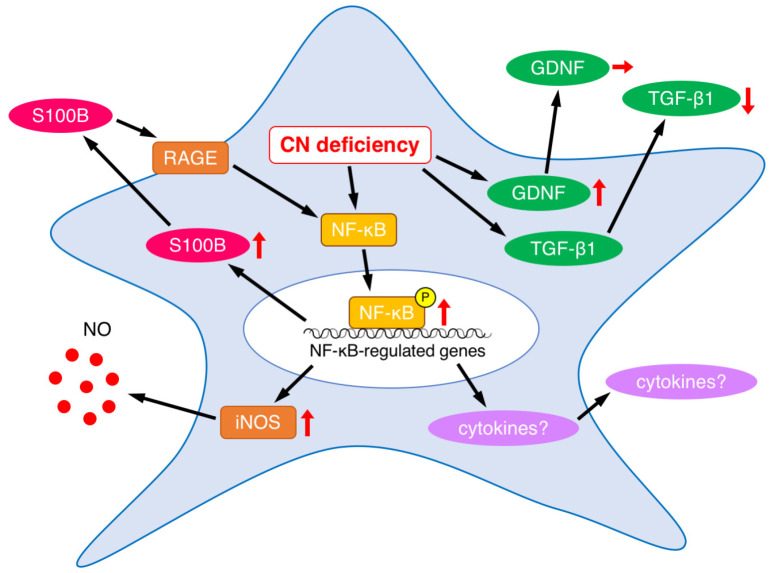
Summary of abnormalities in secreted substances and related proteins in *CNB1*-deficient EGCs. Upward, downward, and rightward arrows represent increase, decrease, and no change, respectively.

## Data Availability

The data presented in this study are available on request from the corresponding author.

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
