# Peer review of "Calcineurin B1 Deficiency Reduces Proliferation, Increases Apoptosis, and Alters Secretion in Enteric Glial Cells of Mouse Small Intestine in Culture"

_cells, 2023, doi:10.3390/cells12141867_

Round 1

Reviewer 1 Report

This study is addressing a relevant topic in the field of enteric glial cells (EGCs) and inflammation of small intestine.  The authors showed that CNB1-deficient EGCs reduced proliferation and increased apoptosis compared with control EGCs. They speculated that reduced proliferation and increased apoptosis of EGCs are potential causes of the dysfunction and inflammation of the small intestine in CNB1-CKO mice.

However, several issues need to be addressed as detailed below. 

1.       All of GFAP positive cells (Whites color) in Figure 2 seem like Hoechst 33342 positive cells (Blue color). Please verify. It would be better if the authors could stain aSMA and GFAP together and get rid of Hoechst33342.

2.       The authors tried to examine the “Effects of S100B Inhibition on CNB1-CKO Mice In Vivo”. However, the authors only showed survival rate and body weight, which cannot enough support the speculation. Please add more information about S100 inhibitor on CNB1-CKO mice in vivo to reduce intestinal inflammation instead of only Body weight, and survival rate, such as disease activity index, tissue length, even H&E staining.

3.       In figure 6, the authors figure out that the NF-kB activation would result in production of various inflammatory cytokines. I am curious why the authors didn’t use cytokine kit or ELISA to investigate further. Please verify it.

Minor points:

1)    There is no labelling in Figure2 B4, C4….

2)    The location of GFAP positive cells in the B4 seem like be moved and doesn’t match B1, B2, B3.

Language can be improved.

Reviewer 2 Report

The authors of the manuscript entitled “Calcineurin B1 deficiency reduces proliferation, increases apoptosis, and alters secretion in enteric glial cells of mouse small intestine” aimed to investigated proliferation,  survival and S100B pathway on the enteric glial cells culture from knockout CNB1 mice. Moreover, the atuhors evaluated the effect of pentamidine on survival and body weight of wild and CKO animals. The authors concluded that CNB1 deficiency leads to reduced proliferation and increased apoptosis of EGCs and abnormal expression and secretion of EGC-derived substances. l. Follow my sugestions and concerns:

Minor review:

At line 206, insert the vehicle of pentamidine isethionate salt.

At figure 2, missing scale bars at representative fotos.

At figure 3, insert molecular weight of the proteins.

Review the figure 5, body weight data should be a Repeated Measures ANOVA. Besides that, unify the B and C graphs.

Major review:

            The authors demonstrated mostly data in enteric glial cells culture. I suggest to the authors in order to improve the manuscript to review the title of the article and especially, add immuno staining of CKO animals.

            The authors perfomed in vivo investigation of pentamidine, an inhibitor of S100B. However during the manuscript, there is no information about dose and references that support the use of this molecule as an inhibitor of S100B. Sigma/ Tocris suggested pentamidine as anti-protozoal and in brain as a NMDA antagonist and inhibitor of constitutive nitric oxide synthase. Clarify this point.

Moreover, at the discussion section, there is no information/discussion about the in vivo data of the S100B inhibitor. I suggest to the authors insert onformation about pentamidine as S100B inhibitor (doi: 10.1016/j.jmb.2008.06.047; doi: 10.3390/ijms12010128; doi: 10.4155/fmc.12.191; doi:10.1016/j.bbrc.2019.07.045; doi: 10.1016/0922-4106(95)90107-8;doi: 10.1186/1742-2094-9-277; doi: 10.1155/2015/508342;  doi: 10.1074/jbc.M405419200; doi: 10.1021/jm0497038). And finally, a conclusion about the in vivo data.

Round 2

Reviewer 1 Report

The authors are addressing that Calcineurin B1 deficiency could reduce proliferation, increases apoptosis, and alter secretion in enteric glial cells of mouse small intestine, which may contribute to dysfunction and inflammation of the small intestine. After the authors revised and responded the comments, the revised manuscript looks much better than before and is starting to be great. No more comments. Hopefully, we could see more data could be presented in the future study. Congratulation!

Reviewer 2 Report

The authors responded to my considerations and suggestions, which improved the article.